

# Effect of PM2.5 on burden of mortality from non-communicable diseases in northern Thailand

Nichapa Parasin[1] and Teerachai Amnuaylojaroen[2,3]

[1] School of Allied Health Science, University of Phayao, Phayao, Thailand
[2] School of Energy and Environment, University of Phayao, Phayao, Thailand
[3] Atmospheric Pollution and Climate Change Research Unit, School of Energy and Environment, University of Phayao, Phayao, Thailand

## ABSTRACT

**Background.** Particulate pollution, especially $PM_{2.5}$ from biomass burning, affects public and human health in northern Thailand during the dry season. Therefore, $PM_{2.5}$ exposure increases non-communicable disease incidence and mortality. This study examined the relationship between $PM_{2.5}$ and NCD mortality, including heart disease, hypertension, chronic lung disease, stroke, and diabetes, in northern Thailand during 2017–2021.

**Methods.** The analysis utilized accurate $PM_{2.5}$ data from the MERRA2 reanalysis, along with ground-based $PM_{2.5}$ measurements from the Pollution Control Department and mortality data from the Division of Non-Communicable Disease, Thailand. The cross-correlation and spearman coefficient were utilized for the time-lag, and direction of the relationship between $PM_{2.5}$ and mortality from NCDs, respectively. The Hazard Quotient (HQ) was used to quantify the health risk of $PM_{2.5}$ to people in northern Thailand.

**Results.** High PM2.5 risk was observed in March, with peak $PM_{2.5}$ concentration reaching 100 µg/m3, with maximum HQ values of $1.78 \pm 0.13$ to $4.25 \pm 0.35$ and $1.45 \pm 0.11$ to $3.46 \pm 0.29$ for males and females, respectively. Hypertension significantly correlated with $PM_{2.5}$ levels, followed by chronic lung disease and diabetes. The cross-correlation analysis showed a strong relationship between hypertansion mortality and $PM_{2.5}$ at a two-year time lag in Chiang Mai (0.73) (CI [−0.43–0.98], $p$-value of 0.0270) and a modest relationship with chronic lung disease at Lampang (0.33) (a four-year time lag). The results from spearman correlation analysis showed that $PM_{2.5}$ concentrations were associated with diabetes mortality in Chiang Mai, with a coefficient of 0.9 (CI [0.09–0.99], $p$-value of 0.03704). Lampang and Phayao had significant associations between PM2.5 and heart disease, with coefficients of 0.97 (CI [0.66–0.99], $p$-value of 0.0048) and 0.90 (CI [0.09–0.99], $p$-value of 0.0374), respectively, whereas Phrae had a high coefficient of 0.99 on stroke.

Corresponding author
Teerachai Amnuaylojaroen,
teerachai4@gmail.com

## INTRODUCTION

$PM_{2.5}$ is a serious air pollutant with a significant influence on public and human health (*Amnuaylojaroen, Parasin & Limsakul, 2022*; *Parasin, Amnuaylojaroen & Saokaew, 2023*). As it rapidly enters the lungs through the respiratory system, it may exacerbate respiratory and mutagenic disorders (*Amoatey, Omidvarborna & Baawain, 2018*). Recently, it has been designated as a Group I carcinogen with serious public health effects by the International Agency for Research on Cancer (*Amnuaylojaroen, Parasin & Limsakul, 2022*; *Amnuaylojaroen & Parasin, 2024*; *Prasannavenkatesh et al., 2015*).

Several studies have shown that ambient $PM_{2.5}$ increases disease incidence and mortality. China and the United States have higher $PM_{2.5}$ related mortality from ischemic heart disease (IHD), stroke, lung cancer, and chronic obstructive pulmonary disease (COPD) (*Tian et al., 2017*; *Chen & Hoek, 2020*; *Pinault et al., 2016*; *Hystad et al., 2020*; *Bowe et al., 2019*). $PM_{2.5}$ exposure has also been linked to diabetes-related mortality (*Feng et al., 2016*; *Bowe et al., 2018*; *Etchie et al., 2017*). Several studies have demonstrated that PM2.5 significantly contributes to the incidence and mortality of various diseases (*Health Effects Institute, 2020*). For instance, in China and the United States, PM2.5-related mortality is notably high for ischemic heart disease (IHD), stroke, lung cancer, and COPD (*Tian et al., 2017*; *Chen & Hoek, 2020*; *Prüss-Ustün et al., 2019*; *Prüss-Ustün et al., 2019*). A recent study found an unambiguous link between $PM_{2.5}$ exposure and NCD mortality. *Atkinson et al. (2014)* found positive correlations between mortality and most other causes of death and cardiovascular and respiratory hospital admissions. Several studies in Thailand, for example, by *Pothirat et al. (2021)* examined the immediate effects of $PM_{2.5}$ on non-accidental mortality and causes of death in Chiang Mai, while *Mueller et al. (2021)* examined the long-term health effects of particle air pollution in Thailand. This study provides insight into $PM_{2.5}$ health risks.

Biomass burning pollutes northern Thailand, especially in January and April (*Yin et al., 2019*; *Amnuaylojaroen et al., 2020*). In addition, air pollution-induced haze is becoming more severe in this region (*Lee et al., 2018*; *Amnuaylojaroen et al., 2014*; *Lee, Iraqui & Wang, 2019*). Several meteorological and topographical factors also exacerbate northern Thailand's air pollution (*Amnuaylojaroen & Kreasuwun, 2012*). $PM_{2.5}$ levels exceed the accepted standard in the dry season (November–April) (*Amnuaylojaroen, Parasin & Limsakul, 2022*; *Parasin, Amnuaylojaroen & Saokaew, 2023*). According to the health risk assessment, all age groups in northern Thailand are at risk from $PM_{2.5}$, especially in February and March (*Amnuaylojaroen & Parasin, 2023a*; *Amnuaylojaroen & Parasin, 2023b*).

Cancers, cardiovascular diseases, diabetes, and chronic respiratory diseases are NCDs caused by physiological, biochemical, behavioral, and environmental factors, particularly air pollution (*Howse et al., 2021*). According to WHO predictions, seven of the top ten killers in 2019 were NCDs. Respiratory and cardiovascular diseases kill most people worldwide (*World Health Organization, 2023*). While air pollution significantly affects morbidity and mortality (*Hoek et al., 2010*). For 545 million people, chronic respiratory disorders were the third leading cause of death in 2017 (*Soriano et al., 2020*).

Several studies have examined the relationship between $PM_{2.5}$ andseveral diseases in Thailand; for example, *Mueller et al. (2021)* and *Pothirat et al. (2021)* studied $PM_{2.5}$ related to health effects in Thailand. *Mueller et al. (2021)* assessed the health effects of prolonged particle air pollution in Thailand. Their study used 1996–2016 data. $PM_{2.5}$ exposure was studied for its health and economic effects on lower respiratory infections (LRIs), stroke, COPD, lung cancer, and ischemic heart disease mortality. Additional studies examined diabetes mortality, dementia, and Parkinson's disease incidence. However, they excluded northern Thailand data due to limited availability. *Pothirat et al. (2021)* examined that particulate matter ($PM_{10}$ and $PM_{2.5}$) affects non-accidental mortality and causes of death from COPD, CAD, and sepsis in Chiang Mai in northern Thailand during 2016 to 2018. Nevertheless, the relationship between $PM_{2.5}$ and NCD mortality in northern Thailand is still poorly understood. This study fills a gap in existing research by examining the effects of $PM_{2.5}$ exposure on NCDs, including heart disease, hypertension, chronic lung disease, stroke, and diabetes, in Thailand's northern provinces, including Chiang Mai, Lamphun, Lampang, Phrae, Nan, Phayao, Chiang Rai, and Mae Hong Son, from 2017 to 2021.

## MATERIALS & METHODS

To examine the relationship between $PM_{2.5}$ concentration and various NCDs in different provinces of northern Thailand, several analyses were conducted. These analyses included spatial analysis, which revealed a possible relationship between higher levels of $PM_{2.5}$ andincreased mortality from NCDs. Time series analysis was also performed to understand the temporal patterns of air pollution and its health impacts. Cross-correlation analysis was used to determine immediate effects, while some effects were found to have a delayed response. Additionally, Spearman correlation analysis was employed to identify specific NCDs that are strongly associated with $PM_{2.5}$.

Between 2017 and 2021, the study employed two datasets to examine the impact of $PM_{2.5}$ on mortality from NCDs in Northern Thailand. This investigation included the first dataset, which was the $PM_{2.5}$ concentration from the Modern-Era Retrospective Analysis for Research and Applications, Version 2 (MERRA-2) reanalysis with resolution of $0.5 \times 0.625$ degrees (*Gelaro et al., 2017*). To account for the limited level of coarse resolution in the MERRA-2 reanalysis data, it is essential to adjust this data using ground-based measurements from eight locations in each province of northern Thailand (Fig. 1, Table S1) obtained from the Pollution Control Department (PCD) throughout the year 2021 which is the years of ground-based measurement data selected for the study were based on the availability of comprehensive datasets, with missing values ranging from 1% to 6% for each province. The precision of ground-based measurement of $PM_{2.5}$ concentration data is closely followed rigorous Quality Assurance and Quality Control (QA/QC) protocols that were based on the guidelines established by the United States Environmental Protection Agency (EPA) (*Sen et al., 2004*). The QA/QC methods included many essential stages. Initially, the monitoring equipment underwent frequent calibration using standard reference materials to guarantee precise results, in accordance with the

manufacturer's instructions and EPA rules. Furthermore, the acquired data were cross-referenced with data from other reputable sources, including satellite observations and adjacent monitoring stations, to verify the precision of the $PM_{2.5}$ readings. The process of cross-referencing aided in the identification and rectification of any inconsistencies or anomalies. Furthermore, any data points that were missing or deviated from the norm were dealt with by using interpolation techniques. These data points were then cross-checked against established trends and patterns to guarantee the dataset's coherence and comprehensiveness. The data collecting techniques were well documented, including precise records of the date, time, and circumstances of each measurement. This process of traceability guaranteed that any deviations or discrepancies could be systematically monitored and examined. Systematic inspections were conducted to detect any irregularities or exceptional values in the data, and appropriate measures were implemented to guarantee the dependability of the dataset. This included the process of re-measuring, if deemed required, or making adjustments to the data using verified correction factors. All individuals participating in data collecting and processing had comprehensive training in quality assurance and quality control methods, as well as in the operation of monitoring equipment. The program director designed and authorized standard operating procedures (SOPs) to ensure consistency and dependability in all data-gathering operations. Regular evaluations and inspections of the quality assurance and quality control processes were carried out to verify adherence to specified standards and pinpoint opportunities for improvement.

The second dataset contains annual mortality for chronic lung diseases, stroke, heart disease, hypertension, and diabetes in the provinces of Chiang Mai, Lamphun, Lampang, Phrae, Nan, Phayao, Chiang Rai, and Mae Hong Son. The Division of Noncommunicable Diseases in Thailand serves as mortality data for NCDs. The study used a dataset including mortality attributed to heart disease, hypertension, chronic lung disease, stroke, and diabetes in the northern region of Thailand spanning the years 2017 to 2021. The Division of Non-Communicable Diseases in Thailand provided mortality data for NCDs. The data on the number of deaths due to various NCDs were sourced from the national health databases maintained by the Ministry of Public Health in Thailand. These databases aggregate mortality data from medical facilities and health authorities countrywide. The relevant mortality data was extracted using specific International Classification of Diseases (ICD) codes related to chronic lung disease, stroke, heart disease, hypertension, and diabetes. In this study, we focused on mortality data rather than morbidity data for non-communicable diseases (NCDs). This decision was driven by several factors: first, mortality data was more consistently available across the study period and regions, providing a robust and comprehensive dataset for analysis. Second, mortality is a definitive and severe outcome that directly reflects the public health burden of PM2.5 exposure, making it a critical measure for assessing the impact of air pollution on health. While morbidity data can provide insights into disease prevalence, the variability and potential underreporting of such data in northern Thailand posed significant challenges. Therefore, we used mortality data to ensure the reliability and validity of our findings.
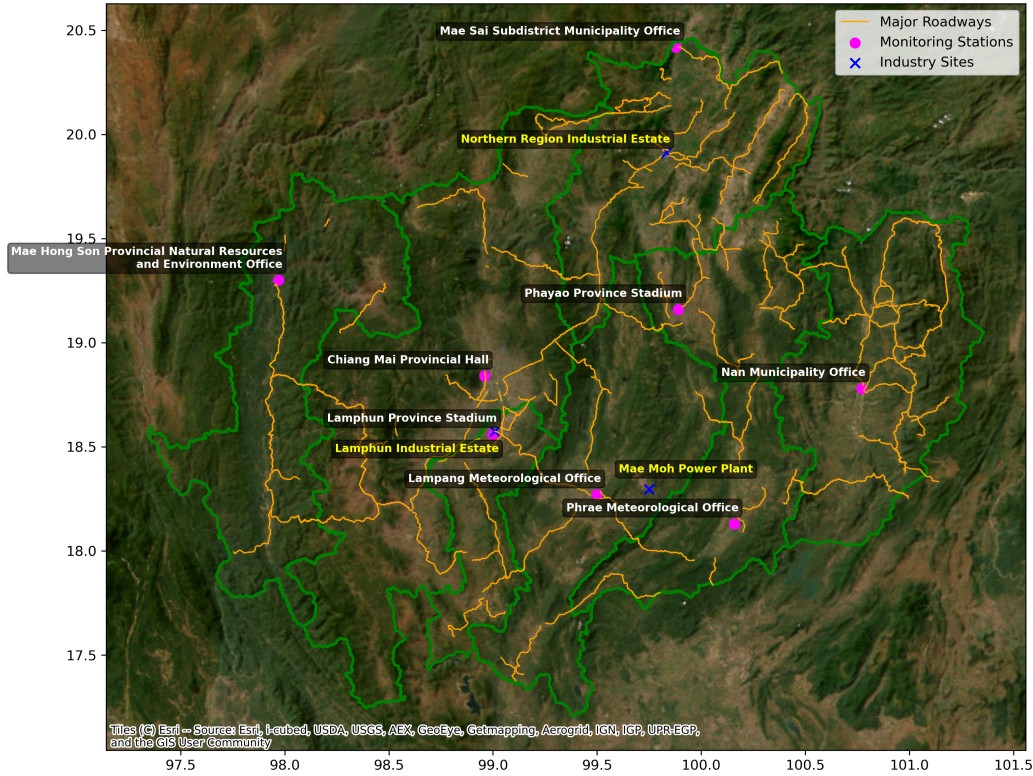

**Figure 1** **Map of northern Thailand demonstrating the monitoring stations (pink circles), major roadways (orange line), and industry sites (blue cross).** Satellite imagery sourced from Esri, i-cubed, USDA, USGS, AEX, GeoEye, Getmapping, Aerogrid, IGN, IGP, UPR-EGP, and the GIS User Community.

To adjust the MERRA2 dataset, the years of 2021 ground-based measurement data from PCD were used to alleviate the existence of missing values to estimate a smoother correction by K-Nearest Neighbors (KNN) Imputation. KNN Imputation is a technique that finds the K-nearest neighbors of a data point with missing values. It uses the Euclidean distance in the feature space to measure how close these neighbors are. After identifying the closest neighbors, the missing values are filled in by calculating the average of the relevant values from these neighbors. This approach guarantees that the inputted values adhere to the established data patterns by using the resemblance between data points (*Troyanskaya et al., 2001*).

The Euclidean distance between two data points $x_i$ and $x_j$ in a $n$-dimensional feature space is computed using the following formula:

$$d\left(x_i, x_j\right) = \sqrt{\sum_{k=1}^{n}(x_{ik} + x_{jk})^2}$$

where $x_i$ and $x_j$ are the $k$-th characteristics of the $i$th and $j$th data points, respectively. The value $\hat{x}_{im}$, which is assigned to a missing feature $m$ in data point $i$, is determined as

follows:

$$\hat{x}_{im} = \frac{1}{k} \sum_{j \in N(i)} x_{jm}$$

$N(i)$ represents the indices of the $k$-nearest neighbors of $i$, whereas $x_{jm}$ refers to the value of feature m in the $j$th neighbor.

We chose a $K$ value of 5 after conducting initial tests that successfully balanced the trade-off between bias and variance. The imputer was built up and used on the dataset, replacing the missing values with estimates obtained from the closest neighbors. By focusing on the overlapping period in 2021, we conducted a linear regression analysis for each monitoring station. The monthly PCD data was used as the dependent variable ($y$), whereas the MERRA-2 data was used as the independent variable ($x$). The linear connection is represented by the following equation:

$$y = \beta_0 + \beta_1 x$$

The symbol $\beta_0$ represents the intercept of the regression line, whereas $\beta$ represents its slope. The coefficients were evaluated using the least squares approach, a technique that minimizes the sum of squared residuals between the observed and predicted values (*Montgomery, Peck & Vining, 2021*).

The regression coefficients derived from the 2021 data were used to determine the MERRA-2 PM$_{2.5}$ values for the whole dataset spanning from 2017 to 2021. The recalculated MERRA-2 values, denoted as $x'$, were determined

$$x' = \beta_0 + \beta_1 x$$

The linear regression correction method enhances the reliability of the MERRA-2 data by aligning it more closely with ground-based observations (*Wilks, 2011*; *Cannon, Sobie & Murdock, 2015*). To evaluate the performance of adjusted PM$_{2.5}$ data from MERRA, several statistical measures including the Mean Absolute Error (MAE), Root Mean Square Error (RMSE), and R-squared (R$^2$) values were applied. The formula of those statistical metrics as follows equations:

$$MAE = \frac{1}{n} \sum_{i=1}^{n} |y_i - \hat{y}_i|$$

$$RMSE = \sqrt{\frac{1}{n} \sum_{i=1}^{n} (y_i - \hat{y}_i)^2}$$

$$R^2 = 1 - \frac{\sum_{i=1}^{n}(y_i - \hat{y}_i)^2}{\sum_{i=1}^{n}(y_i - \bar{y})^2}$$

This study also performed a cross-correlation analysis to investigate possible delayed relationships between PM$_{2.5}$ concentrations and mortality from NCDs. It exposes delayed correlations, when a change in one variable occurs before a change in another variable after a particular period of time ,along with 95% Confident Interval (CI) (*Wong et al.,*
*2001*; *Pope et al., 1995*). The cross-correlation function (CCF) measures the level of similarity between two time series by adjusting the time lag applied to one of them. The annual $PM_{2.5}$ concentration, obtained from modified MERRA2 and NCDs mortality data, was synced for each province and standardized to ensure comparison. Five datasets were created for each province by shifting the original concentration data by 1, 2, 3, and 4 years, resulting in $PM_{2.5}$ data with a temporal lag. The cross-correlation values were calculated between $PM_{2.5}$ concentrations and NCDs mortality for each lag period (varying from 0 to 4 years) using the CCF. The analysis included assessing the patterns of delayed impacts of $PM_{2.5}$ on NCDs mortality by using the highest correlation values. The mathematical definition of the cross-correlation function between two-time series $X(t)$ and $Y(t)$ is:

$$CCF(\tau) = \frac{E[(X(t) - \mu_X)(Y(t + \tau) - \mu_Y)]}{\sigma_X \sigma_Y}$$

where $\tau$ represents the time lag, $E$ denotes the expected value, $\mu_X$ and $\mu_Y$ represent the means of $X(t)$ and $Y(t)$, respectively, and $\sigma_X$ and $\sigma_Y$ indicate their standard deviations. The CCF value, which varies between $-1$ and 1, indicates both the magnitude and direction of the association.

Furthermore, the Spearman correlation analysis (*Spearman, 1987*) was conducted to evaluate the nonlinear associations between $PM_{2.5}$ concentrations and mortality for different NCDs in various provinces. It is a non-parametric measure, meaning it does not assume any specific distribution for the variables. The Spearman correlation coefficient ($\rho$) was then computed as follows this equation:

$$\rho = 1 - \frac{6 \sum d_i^2}{n(n^2 - 1)}$$

where $d_i$ represents the difference between the rankings of each pair of observations, whereas "nnn" represents the total number of observations. The Spearman correlation coefficients were used to ascertain the magnitude and direction of the monotonic association between $PM_{2.5}$ concentrations and NCDs mortality. The study presented a reliable measure of correlation that is less affected by extreme values and non-linear connections, providing a thorough insight into the potential link between changes in air pollution levels and health outcomes in different geographical areas (*Hauke & Kossowski, 2011*).

To evaluate the risk from $PM_{2.5}$ in northern Thailand, Health risk assessment (HIA) is used to assess the potential effects of $PM_{2.5}$ on human health (*Ghaderpoori et al., 2019*). The exposure of human related to air pollutant was described by the average daily dose (ADD) and was calculated as follows Eq. (1)

$$ADD = \frac{C \times IR \times EF \times ED}{BW \times AT} \tag{1}$$

where C is pollutant concentration ($\mu g/m^3$), IR is inhalation rate ($m^3$/day), ED is exposure duration (years), EF is exposure frequency (days/year), AT is averaging exposure time (days), and BW is body weight (kg), These parameters were used the values from previous studies, as shown in Table S2.

While the health risks are described by the HQ, which is the ratio of ADD to reference dose (RfD), was used to determine risk as follows Eq. (2).

$$\text{Hazard Quotient}(\text{HQ}) = \frac{\text{Average Daily Dose}(\text{ADD})\left(\frac{\mu g}{kg} \cdot \text{day}\right)}{\text{Inhalation Reference Dose}(\text{RfD})\left(\frac{\mu g}{kg} \cdot \text{day}\right)}. \tag{2}$$

The inhalation reference dose (RfD) was calculated as follows Eq. (3). Where the valused used for estimation is reduced using the EPA default value (*Hamastia et al., 2019*), namely exposure time (ET) = 24 hours/day, inhalation rate (IR) = 0.83 m$^3$ /hour, body weight (BW) = 70 kg, exposure frequency (EF) = 350 days/year, ED = 30 years, and averaging time (AT) = ED * 365 days/year. While RfC is the inhalation reference concentration refers the safe limit that was proposed by the US-EPA National Ambient Air Quality Standard (NAAQS) in 2006 for PM$_{2.5}$ (namely 35 μg/m$^3$).

$$\text{RfD} = \frac{\text{RfC} \times \text{IR} \times \text{ET} \times \text{EF} \times \text{ED}}{\text{BW} \times \text{AT}}. \tag{3}$$

If HQ is more than 1.0 indicates that there has a risk to sensitive individuals as a result of exposure (*Amnuaylojaroen, Parasin & Limsakul, 2022*), whilst a high chronic risk is denoted for HQ is more than 10 (*Zheng et al., 2016*).

## RESULTS

### Evaluation of corrected PM 5 data

Before analyzing the NCD relationship, we must evaluate the corrected PM$_{2.5}$ data from MERRA2. Figure S1 compares monthly PM$_{2.5}$ concentrations from the Ori-MERRA, Correct-MERRA, and PCD datasets. The adjusted MERRA-2 dataset agrees more with PCD data, especially in peak pollution months like March and April. The correction approach appears to have improved the precision of the MERRA-2 data, making it a more accurate PM$_{2.5}$ concentration estimate. The adjusted MERRA dataset has fewer inconsistencies than the original MERRA-2 data, especially at lower PM$_{2.5}$ concentrations. Table S3 shows how those datasets' assessment metrics validate the correction method. Comparing Correct-MERRA to Ori-MERRA yields a 5.74 MAE, while comparing it to PCD yields 7.82. This suggests that adjusted data is closer to PCD observations. Correct-MERRA has a 5.8 RMSE compared to Ori-MERRA and 9.69 compared to PCD. This indicates better PCD dataset alignment. The R2 values of 0.87 for Correct-MERRA and Ori-MERRA and 0.74 for Correct-MERRA and PCD indicate a robust correlation and improved PM2.5 concentration estimation after the adjustment.

### PM$_{2.5}$ in northern Thailand

Figure S2 shows PM$_{2.5}$ air quality in northern Thailand. It shows average monthly PM$_{2.5}$ concentrations in eight Northern Thai provinces from 2017 to 2021. PM$_{2.5}$ concentrations peaked in February, March, and April during the dry season. PM$_{2.5}$ concentrations are highest in Chiang Mai and Mae Hong Son, peaking at 100 μg/m$^3$ in March. Peak levels exceed WHO Annual and 24-hour Standards of 5 and 15 μg/m$^3$, as well as Thailand Annual and 24-hour Standards of 15 and 37.5 μg/m$^3$, respectively. These criteria are

disregarded during peak months, emphasizing the dry season air quality issues in these areas. In contrast, all provinces have lower $PM_{2.5}$ concentrations during the rainy season (June–September). Enhanced precipitation serves to eliminate particulate matter from the atmosphere, lowering $PM_{2.5}$ levels at this time of year. However, even in these months, some provinces still meet or exceed the WHO's annual standard, indicating the area's long-standing air pollution problem.

## Health risk assessment

Figure S3 shows the monthly average HQ for $PM_{2.5}$ exposure across genders in northern Thailand during 2017–2021. The monthly average HQ values exceeded one in most provinces from February to April, indicating significant risks. As shown in Table 1, the monthly mean HQ of adult males at Chiang Mai displays a mean HQ of 1.06 ±0.04. The values in Chiang Rai, Lampang, Lamphun, Nan, Phayao, Mae Hong Son, and Phrae are 1.14 ± 0.04, 0.82 ± 0.02, 0.95 ± 0.02, 0.94 ± 0.02, 0.99 ± 0.02, 1.00 ± 0.02, and 1.05 ± 0.02 respectively. While the monthly mean HQ of adult females at Chiang Mai has a mean HQ of 0.85 ± 0.03. The values in Chiang Rai, Lampang, Lamphun, Nan, Phayao, Mae Hong Son, and Phrae are 0.93 ± 0.04, 0.67 ± 0.02, 0.77 ± 0.03, 0.76 ± 0.02, 0.80 ± 0.01, 0.81 ± 0.02, and 0.86 ± 0.19, respectively.

## Effect of $PM_{2.5}$ on NCDs

Figure 2 displays the annual levels of $PM_{2.5}$ concentrations and the mortality associated with five NCDs in different provinces between 2017 and 2021. Figure 2A displays the annual $PM_{2.5}$ concentration, while Figs. 2B to 2F depict mortality for heart disease, hypertension, chronic lung disease, stroke, and diabetes, respectively. The province of Chiang Mai had the greatest concentration of $PM_{2.5}$ , peaking at 27 $\mu g/m^3$, and Lampang had the lowest value at 20 $\mu g/m^3$. Chiang Mai has the highest mortality across all five NCDs. Specifically, there were 460.3 deaths due to heart disease, 386.4 deaths due to hypertension, 360.2 deaths due to chronic lung disease, 842.8 deaths due to stroke, and 338.8 deaths due to diabetes.

Figure 3 shows $PM_{2.5}$ concentrations and NCD mortality in Thailand's provinces from 2017 to 2021. $PM_{2.5}$ concentrations in Chiang Mai peak in 2019 and then decline. Stroke mortality has increased, especially after 2019. Time trends for other NCDs are consistent or decreasing. $PM_{2.5}$ concentrations in Lamphun increased in 2019, triggering stroke mortality. The other NCDs in Lamphun showed mixed trends and modest changes. $PM_{2.5}$ concentrations in Lampang increased significantly in 2019, then decreased. Although stroke mortality is rising, other NCDs remain stable with slight variations. In 2019, $PM_{2.5}$ concentrations increased in Phrae (Fig. 3D) before decreasing. Other NCDs have trends with minor fluctuations, but stroke mortality rises gradually. Nan (Fig. 3E) had the highest $PM_{2.5}$ concentrations in 2019 and then decreased. Stroke and heart disease mortality rise, but other NCDs remain stable or vary slightly. $PM_{2.5}$ concentrations in Phayao (Fig. 3F) increased in 2019 and then decreased. Stroke mortality rises, but other NCDs show small differences. In Chiang Rai (Fig. 3G), $PM_{2.5}$ concentrations increased in 2019 and then decreased. While stroke mortality rises, other NCDs remain stable with

Parasin and Amnuaylojaroen (2024), *PeerJ*, DOI 10.7717/peerj.18055

**Table 1  Monthly means of the HQ related to PM$_{2.5}$ according to male and female in eight provinces during 2017–2021.**

| Month | Chiang Mai | | Chiangrai | | Lampang | | Lamphun | | Nan | | Phayao | | Mae Hong Son | | Phrae | |
|---|---|---|---|---|---|---|---|---|---|---|---|---|---|---|---|---|
| | Male | Female | Male | Female | Male | Female | Male | Female | Male | Female | Male | Female | Male | Female | Male | Female |
| January | 1.17 ± 0.08 | 0.95 ± 0.08 | 0.81 ± 0.05 | 0.66 ± 0.05 | 1.32 ± 0.17 | 1.07 ± 0.14 | 1.28 ± 0.20 | 1.04 ± 0.16 | 1.03 ± 0.20 | 0.84 ± 0.16 | 1.20 ± 0.22 | 0.98 ± 0.18 | 0.98 ± 0.39 | 0.80 ± 0.32 | 1.50 ± 0.24 | 1.22 ± 0.19 |
| February | 1.71 ± 0.11 | 1.39 ± 0.09 | 1.35 ± 0.12 | 1.10 ± 0.10 | 1.73 ± 0.17 | 1.40 ± 0.14 | 1.72 ± 0.24 | 1.40 ± 0.19 | 1.64 ± 0.24 | 1.34 ± 0.19 | 1.67 ± 0.24 | 1.36 ± 0.19 | 2.60 ± 0.39 | 2.11 ± 0.32 | 1.99 ± 0.24 | 1.62 ± 0.19 |
| March | 2.70 ± 0.18 | 2.20 ± 0.15 | 4.25 ± 0.35 | 3.46 ± 0.29 | 1.78 ± 0.13 | 1.45 ± 0.11 | 2.35 ± 0.33 | 1.91 ± 0.27 | 2.56 ± 0.35 | 2.08 ± 0.29 | 3.08 ± 0.35 | 2.51 ± 0.29 | 4.48 ± 0.39 | 3.65 ± 0.32 | 2.28 ± 0.24 | 1.86 ± 0.19 |
| April | 1.86 ± 0.12 | 1.51 ± 0.10 | 2.90 ± 0.23 | 2.36 ± 0.18 | 1.42 ± 0.11 | 1.16 ± 0.09 | 1.68 ± 0.24 | 1.37 ± 0.19 | 1.72 ± 0.22 | 1.40 ± 0.18 | 1.61 ± 0.24 | 1.31 ± 0.19 | 2.17 ± 0.39 | 1.77 ± 0.32 | 1.35 ± 0.24 | 1.10 ± 0.19 |
| May | 0.84 ± 0.06 | 0.68 ± 0.05 | 1.25 ± 0.10 | 1.01 ± 0.08 | 0.76 ± 0.06 | 0.62 ± 0.05 | 0.80 ± 0.08 | 0.65 ± 0.07 | 0.94 ± 0.07 | 0.76 ± 0.06 | 0.66 ± 0.07 | 0.53 ± 0.06 | 0.54 ± 0.39 | 0.44 ± 0.32 | 0.83 ± 0.24 | 0.68 ± 0.19 |
| June | 0.49 ± 0.02 | 0.40 ± 0.02 | 0.36 ± 0.01 | 0.29 ± 0.01 | 0.32 ± 0.01 | 0.26 ± 0.01 | 0.33 ± 0.02 | 0.26 ± 0.01 | 0.35 ± 0.02 | 0.29 ± 0.01 | 0.27 ± 0.02 | 0.22 ± 0.01 | 0.17 ± 0.02 | 0.14 ± 0.01 | 0.30 ± 0.24 | 0.25 ± 0.19 |
| July | 0.47 ± 0.02 | 0.38 ± 0.02 | 0.28 ± 0.01 | 0.23 ± 0.01 | 0.34 ± 0.02 | 0.27 ± 0.02 | 0.37 ± 0.02 | 0.30 ± 0.01 | 0.36 ± 0.02 | 0.29 ± 0.01 | 0.24 ± 0.02 | 0.19 ± 0.01 | 0.13 ± 0.02 | 0.10 ± 0.01 | 0.27 ± 0.24 | 0.22 ± 0.19 |
| August | 0.50 ± 0.02 | 0.41 ± 0.02 | 0.29 ± 0.01 | 0.24 ± 0.01 | 0.35 ± 0.02 | 0.28 ± 0.02 | 0.42 ± 0.02 | 0.34 ± 0.02 | 0.35 ± 0.02 | 0.28 ± 0.01 | 0.24 ± 0.02 | 0.20 ± 0.01 | 0.16 ± 0.02 | 0.13 ± 0.01 | 0.28 ± 0.24 | 0.23 ± 0.19 |
| September | 0.52 ± 0.02 | 0.42 ± 0.02 | 0.37 ± 0.02 | 0.30 ± 0.02 | 0.39 ± 0.02 | 0.32 ± 0.02 | 0.48 ± 0.02 | 0.39 ± 0.02 | 0.40 ± 0.02 | 0.32 ± 0.02 | 0.42 ± 0.02 | 0.34 ± 0.01 | 0.17 ± 0.02 | 0.14 ± 0.01 | 0.44 ± 0.24 | 0.36 ± 0.19 |
| October | 0.60 ± 0.02 | 0.48 ± 0.02 | 0.41 ± 0.02 | 0.33 ± 0.02 | 0.30 ± 0.02 | 0.24 ± 0.02 | 0.61 ± 0.02 | 0.49 ± 0.02 | 0.48 ± 0.02 | 0.39 ± 0.02 | 0.45 ± 0.02 | 0.37 ± 0.01 | 0.26 ± 0.02 | 0.21 ± 0.01 | 0.45 ± 0.24 | 0.36 ± 0.19 |
| November | 0.78 ± 0.02 | 0.63 ± 0.02 | 0.59 ± 0.02 | 0.48 ± 0.02 | 0.39 ± 0.02 | 0.32 ± 0.02 | 0.83 ± 0.02 | 0.67 ± 0.02 | 0.61 ± 0.02 | 0.49 ± 0.02 | 0.67 ± 0.02 | 0.55 ± 0.01 | 0.39 ± 0.02 | 0.32 ± 0.01 | 0.69 ± 0.24 | 0.56 ± 0.19 |
| December | 0.98 ± 0.04 | 0.80 ± 0.04 | 0.85 ± 0.04 | 0.70 ± 0.04 | 0.72 ± 0.02 | 0.59 ± 0.02 | 1.09 ± 0.05 | 0.89 ± 0.05 | 0.84 ± 0.02 | 0.68 ± 0.02 | 1.30 ± 0.02 | 1.06 ± 0.01 | 0.6 ± 0.02 | 0.49 ± 0.01 | 1.01 ± 0.20 | 0.82 ± 0.20 |
| Mean | 1.06 ± 0.04 | 0.85 ± 0.03 | 1.14 ± 0.04 | 0.93 ± 0.04 | 0.82 ± 0.02 | 0.67 ± 0.02 | 0.95 ± 0.02 | 0.77 ± 0.03 | 0.94 ± 0.02 | 0.76 ± 0.02 | 0.99 ± 0.02 | 0.80 ± 0.01 | 1.00 ± 0.02 | 0.81 ± 0.02 | 1.05 ± 0.02 | 0.86 ± 0.19 |

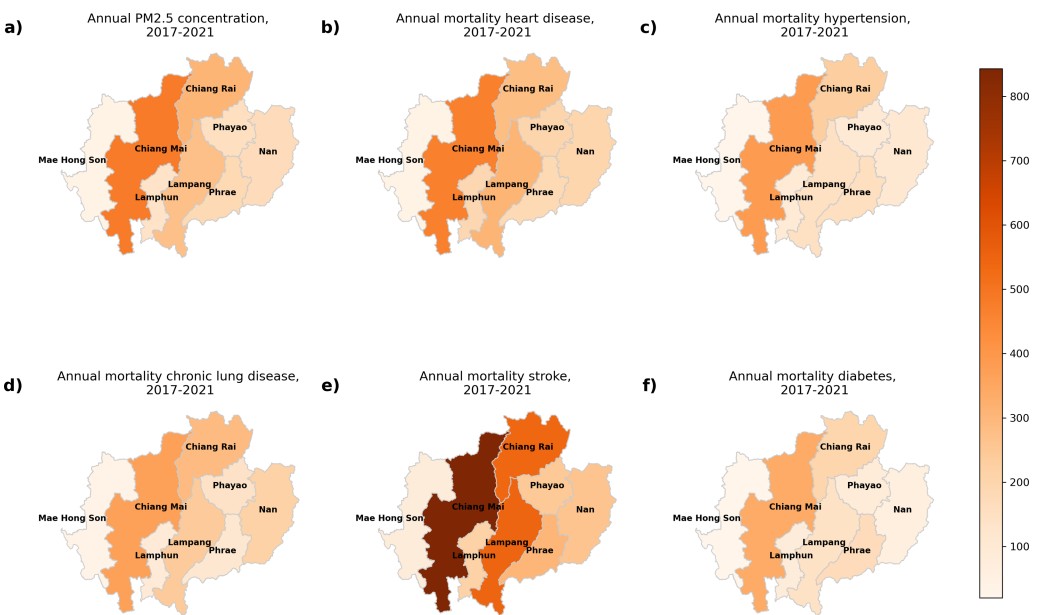

**Figure 2** The annual of (A) PM$_{2.5}$ concentration, and mortality from (B) heart disease, (C) hypertension, (D) chronic lung disease, (E) stroke, and (F) diabetes during 2017–2021 in northern Thailand. Map imagery sourced from Bing, GeoNames, Microsoft, TomTom.

slight variations. PM2.5 concentrations in Mae Hong Son (Fig. 3H) peaked in 2019 and then decreased. Stroke mortality is rising, while other NCDs show mixed trends with small differences.

Figure 4A and Table 2 show chronic lung disease mortality had the highest negative correlation of −0.64 at a lag of zero in Chiang Mai, suggesting an immediate effect. A significant association between hypertension mortality and a two-year delay (correlation coefficient 0.73, CI [−0.43 to 0.98], *p*-value of 0.0270) suggests a delayed impact. Stroke and heart disease mortality have no statistically significant delayed effects in Chiang Mai. Figure 4B and Table 2 show that Chiang Rai's influence on most NCDs mortality varies and does not have a consistent lag time. The correlation between stroke mortality and zero lag time delay is 0.02 (CI [−0.88 to 0.89], *p*-value of 0.97), indicating no significant association. A statistically significant correlation (0.33) (CI [−0.78 to 0.94], *p*-value of 0.0410) exists between chronic lung disease mortality and a four-year delay in Lampang. This suggests a long-term impact, as shown in Fig. 4C and Table 2. Unlike other diseases, hypertension and diabetes mortality have no significant correlation, suggesting different effects. A correlation coefficient of 0.82 at zero lag (CI [−0.22 to 0.99]), *p*-value of 0.782) indicates a significant and rapid impact of PM$_{2.5}$ on stroke mortality in Phrae. PM$_{2.5}$ does not strongly correlate with other NCDs such heart disease or diabetes (Fig. 4D, Table 2). There is a significant inverse relationship between Nan and stroke mortality, with a four-year lag (−0.44, CI [−0.95 to 0.72], *p*-value 0.0080) (Fig. 4E, Table 2). As shown in Fig. 4F and Table 2, Phayao shows inconsistent patterns with no significant connections for most NCDs, suggesting that PM$_{2.5}$ and NCDs mortality may be affected by other

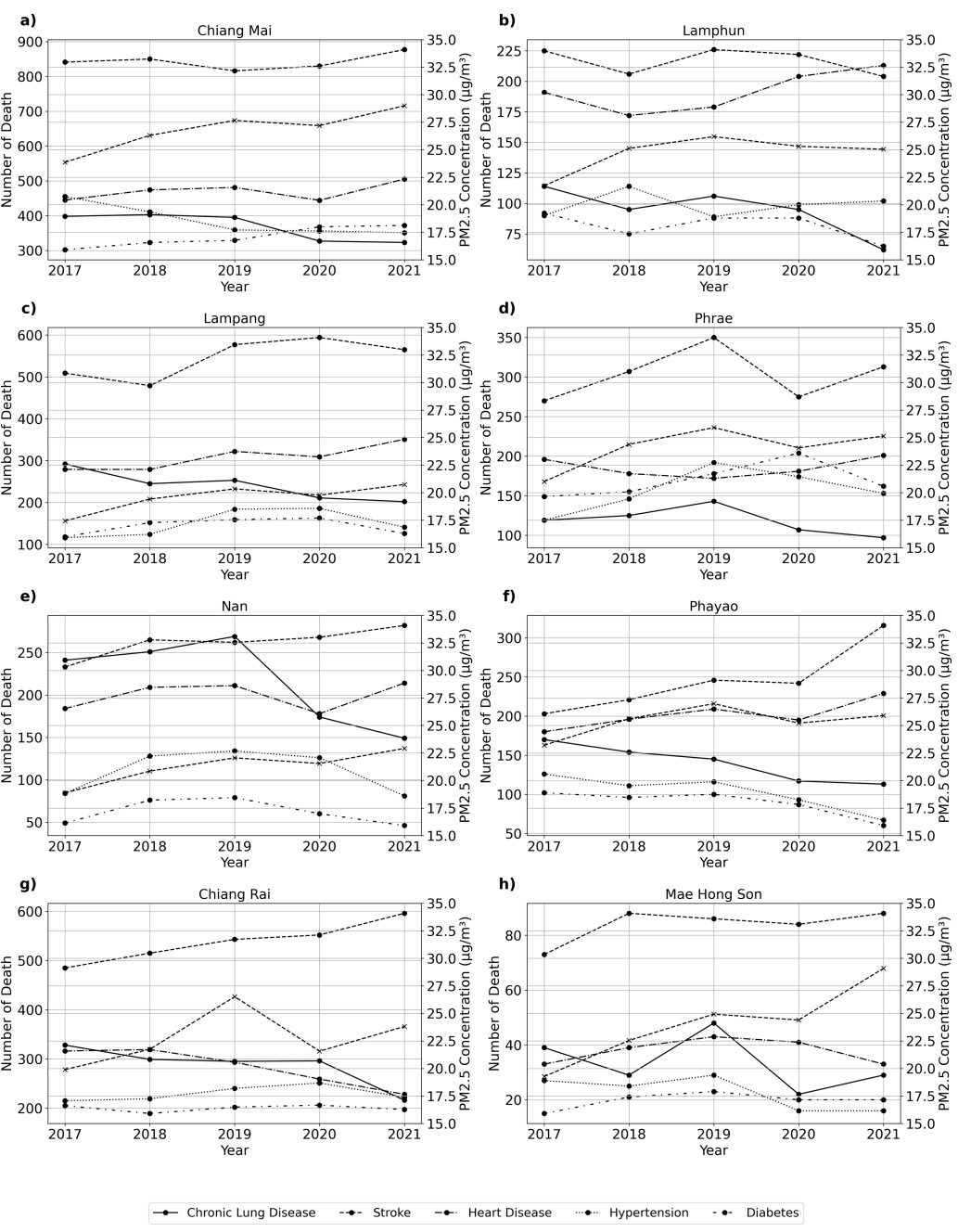

**Figure 3** Time series of PM$_{2.5}$ concentration and number of death from various NCDs at (A) Chiang Mai, (B) Chiang Rai, (C) Lampang, (D) Phrae, (D) Nan, (F) Phayao, (G) Lamphun, and (H) Mae Hong Son.

factors. The association between chronic lung disease and a zero-lag is $-0.48$ (CI [$-0.96$ to 0.70], $p$-value of 0.714), indicating no immediate significant effect. PM$_{2.5}$ levels and NCD mortality are not significantly associated in Lamphun. Diabetes mortality and PM$_{2.5}$ concentrations have a strong inverse relationship in Mae Hong Son, with a correlation

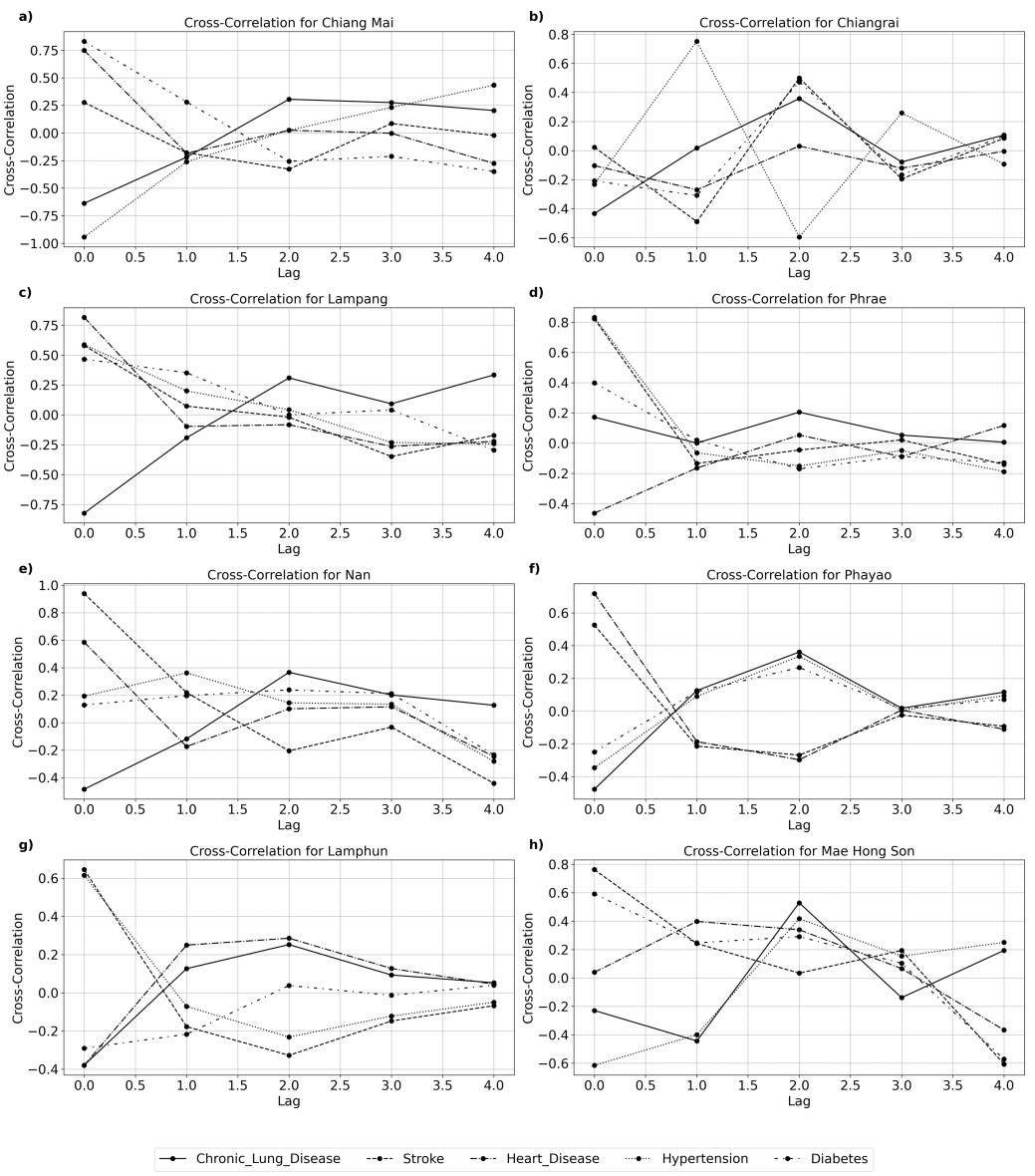

**Figure 4** Cross-correlation between PM$_{2.5}$ concentration and variuos NCD at (A) Chiang Mai, (B) Chiang Rai, (C) Lampang, (D) Phrae, (D) Nan, (F) Phayao, (G) Lamphun, and (H) Mae Hong Son.

coefficient of −0.50 (CI [−0.96 to 0.68], *p*-value of 0.0440). This suggests that PM$_{2.5}$ levels affect diabetes later on.

The Spearman correlation analysis, as shown in Table 3, examines the relationship between PM$_{2.5}$ concentrations and several NCDs in eight provinces in Northern Thailand. There is a strong association between PM$_{2.5}$ levels and the morality of hypertension and diabetes in Chiang Mai. The data shows that there is a strong negative correlation of −0.9 (95% CI [−0.99 to 0.09], *p*-value of 0.0374) between hypertension and PM$_{2.5}$ concentrations. This indicates that when PM$_{2.5}$ levels rise, the number of hypertension

**Table 2  Cross-correlation analysis between PM$_{2.5}$ concentrations and various NCDs in across Northern Thailand.** Cross-correlation values, 95% confidence interval (CI), and *p*-values are presented for each lag from lag0 to lag4.

| Province | Lag | Chronic Lung Diseaase | | Stroke | | Hearth Disease | | Hypertension | | Diabetes | |
|---|---|---|---|---|---|---|---|---|---|---|---|
| | | Cross-Correlation (95% CI) | *P*-value | Cross-Correlation (95% CI) | *P*-value | Cross-Correlation (95% CI) | *P*-value | Cross-Correlation (95% CI) | *P*-value | Cross-Correlation (95% CI) | *P*-value |
| Chiang Mai | 0 | −0.64 (−0.97, 0.56) | 0.602 | 0.28 (−0.80, 0.93) | 0.887 | 0.75 (−0.39, 0.98) | 0.598 | −0.62 (−0.97, 0.58) | 0.596 | 0.21 (−0.83, 0.92) | 0.724 |
| | 1 | −0.22 (−0.92, 0.82) | 0.646 | −0.18 (−0.92, 0.84) | 0.709 | −0.18 (−0.92, 0.83) | 0.693 | −0.01 (−0.89, 0.88) | 0.975 | −0.20 (−0.92, 0.83) | 0.723 |
| | 2 | 0.31 (−0.79, 0.94) | 0.397 | −0.33 (−0.94, 0.78) | 0.396 | 0.02 (−0.88, 0.89) | 0.937 | **0.73 (−0.43, 0.98)** | **0.0270\*** | −0.48 (−0.96, 0.70) | 0.221 |
| | 3 | 0.28 (−0.80, 0.93) | 0.349 | 0.09 (−0.86, 0.90) | 0.74 | 0.00 (−0.88, 0.88) | 0.984 | 0.14 (−0.85, 0.91) | 0.6 | −0.10 (−0.90, 0.86) | 0.685 |
| | 4 | 0.20 (−0.83, 0.92) | 0.245 | −0.02 (−0.89, 0.88) | 0.808 | −0.28 (−0.93, 0.80) | 0.157 | −0.22 (−0.92, 0.82) | 0.179 | 0.18 (−0.84, 0.92) | 0.359 |
| Chiangrai | 0 | −0.43 (−0.95, 0.73) | 0.704 | 0.02 (−0.88, 0.89) | 0.97 | −0.10 (−0.90, 0.86) | 0.83 | −0.09 (−0.90, 0.86) | 0.831 | 0.17 (−0.84, 0.92) | 0.784 |
| | 1 | 0.02 (−0.88, 0.89) | 0.972 | −0.49 (−0.96, 0.69) | 0.293 | −0.27 (−0.93, 0.80) | 0.609 | 0.48 (−0.70, 0.96) | 0.296 | −0.25 (−0.93, 0.81) | 0.622 |
| | 2 | 0.36 (−0.77, 0.94) | 0.332 | 0.50 (−0.68, 0.96) | 0.201 | 0.03 (−0.88, 0.89) | 0.933 | −0.11 (−0.90, 0.86) | 0.761 | 0.23 (−0.82, 0.92) | 0.556 |
| | 3 | −0.08 (−0.90, 0.86) | 0.749 | −0.19 (−0.92, 0.83) | 0.485 | −0.12 (−0.91, 0.85) | 0.704 | −0.27 (−0.93, 0.80) | 0.354 | 0.13 (−0.85, 0.91) | 0.66 |
| | 4 | 0.11 (−0.86, 0.90) | 0.483 | 0.08 (−0.86, 0.90) | 0.643 | −0.01 (−0.88, 0.88) | 0.963 | 0.14 (−0.85, 0.91) | 0.433 | −0.15 (−0.91, 0.85) | 0.41 |
| Lampang | 0 | −0.82 (−0.99, 0.22) | 0.604 | 0.58 (−0.62, 0.97) | 0.696 | 0.81 (−0.24, 0.99) | 0.748 | 0.20 (−0.83, 0.92) | 0.677 | 0.79 (−0.31, 0.99) | 0.739 |
| | 1 | −0.19 (−0.92, 0.83) | 0.645 | 0.07 (−0.87, 0.90) | 0.866 | −0.10 (−0.90, 0.86) | 0.801 | −0.53 (−0.96, 0.66) | 0.214 | −0.56 (−0.97, 0.64) | 0.138 |
| | 2 | 0.31 (−0.79, 0.94) | 0.361 | −0.02 (−0.89, 0.88) | 0.957 | −0.08 (−0.90, 0.86) | 0.832 | −0.34 (−0.94, 0.78) | 0.39 | −0.18 (−0.92, 0.84) | 0.675 |
| | 3 | 0.09 (−0.86, 0.90) | 0.733 | −0.35 (−0.94, 0.77) | 0.229 | −0.26 (−0.93, 0.81) | 0.392 | 0.08 (−0.86, 0.90) | 0.809 | −0.21 (−0.92, 0.83) | 0.477 |
| | 4 | **0.33 (−0.78, 0.94)** | **0.0410\*** | −0.17 (−0.92, 0.84) | 0.374 | −0.22 (−0.92, 0.82) | 0.199 | 0.22 (−0.82, 0.92) | 0.244 | 0.27 (−0.80, 0.93) | 0.13 |
| Phrae | 0 | 0.17 (−0.84, 0.92) | 0.816 | 0.82 (−0.22, 0.99) | 0.782 | −0.46 (−0.96, 0.71) | 0.573 | 0.26 (−0.81, 0.93) | 0.762 | 0.12 (−0.85, 0.91) | 0.786 |
| | 1 | 0.00 (−0.88, 0.88) | 1 | −0.13 (−0.91, 0.85) | 0.734 | −0.16 (−0.91, 0.84) | 0.729 | −0.58 (−0.97, 0.62) | 0.201 | −0.63 (−0.97, 0.57) | 0.136 |
| | 2 | 0.21 (−0.83, 0.92) | 0.598 | −0.05 (−0.89, 0.87) | 0.892 | 0.05 (−0.87, 0.89) | 0.897 | −0.26 (−0.93, 0.81) | 0.526 | −0.11 (−0.90, 0.86) | 0.758 |
| | 3 | 0.05 (−0.87, 0.89) | 0.854 | 0.02 (−0.88, 0.89) | 0.919 | −0.09 (−0.90, 0.86) | 0.737 | 0.01 (−0.88, 0.88) | 0.958 | 0.08 (−0.86, 0.90) | 0.785 |
| | 4 | 0.01 (−0.88, 0.88) | 0.939 | −0.14 (−0.91, 0.85) | 0.433 | 0.12 (−0.85, 0.91) | 0.538 | 0.25 (−0.81, 0.93) | 0.212 | 0.17 (−0.84, 0.91) | 0.382 |
| Nan | 0 | −0.48 (−0.96, 0.69) | 0.64 | 0.94 (0.33, 1.00) | 0.56 | 0.59 (−0.61, 0.97) | 0.577 | 0.77 (−0.35, 0.98) | 0.708 | 0.76 (−0.38, 0.98) | 0.622 |
| | 1 | −0.12 (−0.91, 0.85) | 0.806 | 0.22 (−0.82, 0.92) | 0.59 | −0.17 (−0.92, 0.84) | 0.69 | −0.30 (−0.94, 0.79) | 0.494 | −0.16 (−0.91, 0.84) | 0.721 |
| | 2 | 0.37 (−0.76, 0.94) | 0.334 | −0.21 (−0.92, 0.83) | 0.547 | 0.10 (−0.86, 0.90) | 0.787 | −0.09 (−0.90, 0.86) | 0.819 | −0.09 (−0.90, 0.86) | 0.823 |
| | 3 | 0.20 (−0.83, 0.92) | 0.473 | −0.03 (−0.89, 0.87) | 0.896 | 0.12 (−0.85, 0.91) | 0.664 | −0.11 (−0.90, 0.85) | 0.695 | −0.12 (−0.91, 0.85) | 0.675 |
| | 4 | 0.13 (−0.85, 0.91) | 0.468 | **−0.44 (−0.95, 0.72)** | **0.0080\*** | −0.25 (−0.93, 0.81) | 0.199 | 0.09 (−0.86, 0.90) | 0.653 | 0.07 (−0.87, 0.90) | 0.607 |
| Phayao | 0 | −0.48 (−0.96, 0.70) | 0.714 | 0.53 (−0.67, 0.96) | 0.702 | 0.72 (−0.45, 0.98) | 0.727 | −0.47 (−0.96, 0.71) | 0.66 | −0.37 (−0.94, 0.76) | 0.72 |
| | 1 | 0.13 (−0.85, 0.91) | 0.771 | −0.21 (−0.92, 0.82) | 0.625 | −0.19 (−0.92, 0.83) | 0.658 | 0.16 (−0.84, 0.91) | 0.755 | 0.14 (−0.85, 0.91) | 0.745 |
| | 2 | 0.36 (−0.77, 0.94) | 0.291 | −0.27 (−0.93, 0.80) | 0.466 | −0.30 (−0.93, 0.79) | 0.406 | −0.22 (−0.92, 0.82) | 0.591 | −0.11 (−0.91, 0.85) | 0.767 |
| | 3 | 0.02 (−0.88, 0.89) | 0.938 | −0.02 (−0.89, 0.88) | 0.916 | 0.01 (−0.88, 0.88) | 0.98 | 0.33 (−0.78, 0.94) | 0.263 | 0.25 (−0.81, 0.93) | 0.41 |
| | 4 | 0.12 (−0.85, 0.91) | 0.502 | −0.09 (−0.90, 0.86) | 0.547 | −0.11 (−0.90, 0.86) | 0.472 | 0.04 (−0.87, 0.89) | 0.782 | 0.03 (−0.88, 0.89) | 0.817 |

**Table 2** (*continued*)

| Province | Lag | Chronic Lung Diseaase | | Stroke | | Hearth Disease | | Hypertension | | Diabets | |
|---|---|---|---|---|---|---|---|---|---|---|---|
| | | Cross-Correlation (95% CI) | P-value | Cross-Correlation (95% CI) | P-value | Cross-Correlation (95% CI) | P-value | Cross-Correlation (95% CI) | P-value | Cross-Correlation (95% CI) | P-value |
| Lamphun | 0 | −0.38 (−0.95, 0.76) | 0.875 | 0.64 (−0.55, 0.97) | 0.676 | −0.38 (−0.95, 0.76) | 0.684 | 0.58 (−0.62, 0.97) | 0.559 | −0.11 (−0.90, 0.86) | 0.811 |
| | 1 | 0.13 (−0.85, 0.91) | 0.739 | −0.18 (−0.92, 0.84) | 0.708 | 0.25 (−0.81, 0.93) | 0.636 | −0.55 (−0.96, 0.65) | 0.216 | −0.67 (−0.98, 0.52) | 0.154 |
| | 2 | 0.25 (−0.81, 0.93) | 0.493 | −0.33 (−0.94, 0.78) | 0.36 | 0.29 (−0.80, 0.93) | 0.403 | −0.48 (−0.96, 0.70) | 0.209 | 0.19 (−0.83, 0.92) | 0.639 |
| | 3 | 0.09 (−0.86, 0.90) | 0.729 | −0.15 (−0.91, 0.84) | 0.582 | 0.13 (−0.85, 0.91) | 0.611 | 0.10 (−0.86, 0.90) | 0.692 | 0.29 (−0.80, 0.93) | 0.308 |
| | 4 | 0.05 (−0.87, 0.89) | 0.693 | −0.07 (−0.90, 0.87) | 0.656 | 0.04 (−0.87, 0.89) | 0.816 | 0.17 (−0.84, 0.91) | 0.372 | −0.13 (−0.91, 0.85) | 0.514 |
| Mae Hong Son | 0 | −0.23 (−0.92, 0.82) | 0.594 | 0.76 (−0.37, 0.98) | 0.657 | 0.04 (−0.87, 0.89) | 0.916 | −0.78 (−0.99, 0.32) | 0.697 | 0.53 (−0.66, 0.96) | 0.88 |
| | 1 | −0.44 (−0.95, 0.72) | 0.37 | 0.24 (−0.81, 0.93) | 0.587 | 0.40 (−0.75, 0.95) | 0.358 | 0.18 (−0.83, 0.92) | 0.664 | 0.43 (−0.73, 0.95) | 0.321 |
| | 2 | 0.53 (−0.66, 0.96) | 0.175 | 0.03 (−0.87, 0.89) | 0.931 | 0.34 (−0.77, 0.94) | 0.348 | 0.21 (−0.83, 0.92) | 0.582 | 0.33 (−0.78, 0.94) | 0.334 |
| | 3 | −0.14 (−0.91, 0.85) | 0.638 | 0.19 (−0.83, 0.92) | 0.455 | 0.07 (−0.87, 0.90) | 0.814 | 0.28 (−0.80, 0.93) | 0.342 | **-0.50 (−0.96, 0.68)** | **0.0440*** |
| | 4 | 0.19 (−0.83, 0.92) | 0.367 | **-0.61 (−0.97, 0.59)** | **0.0050*** | −0.37 (−0.94, 0.76) | 0.054 | 0.08 (−0.86, 0.90) | 0.656 | −0.17 (−0.92, 0.84) | 0.299 |

**Notes.**
*Values in bold with an asterisk (*) indicate statistically significant results ($p < 0.05$).

mortality also tends to increase. Similarly, there is a strong positive association of 0.9 (95% CI [0.09–0.99], *p*-value of 0.0374) between diabetes and PM2.5 concentrations, suggesting that greater levels of $PM_{2.5}$ are associated with a higher mortality of diabetes. Chronic lung disease, stroke, and heart disease, among other NCDs, do not exhibit substantial correlations, indicating a lack of strong association with $PM_{2.5}$ in this region. There are no significant connections between $PM_{2.5}$ concentrations and any of NCDs in Chiangrai. Chronic lung disease, stroke, heart disease, hypertension, and diabetes all had *p*-values of 0.1, suggesting weak or non-significant associations. Lampang has a significant positive relationship between PM2.5 levels and the mortality of heart disease, as shown by a correlation coefficient of 0.97 (95% CI [0.66–0.99], *p*-value of 0.0048). This is a substantial correlation between increased $PM_{2.5}$ concentrations and an increase in instances of heart disease. There are no significant associations observed with other disorders such as chronic lung disease, stroke, hypertension, and diabetes. A high relationship (correlation coefficient of 1.0) is detected between $PM_{2.5}$ andstroke in Phrae. The 95% CI for this correlation is 1.00 to 1.00, with a *p*-value of 0.0000, suggesting an extraordinarily strong relationship. There are no other non-communicable diseases in Phrae that have notable relationships. Nan exhibits no notable associations between $PM_{2.5}$ concentrations and any of the non-communicable diseases (NCDs), such as chronic lung disease, stroke, heart disease, hypertension, and diabetes. All *p*-values above 0.1, suggesting the presence of weak relationships. Phayao shows a strong positive relationship with heart disease (correlation coefficient of 0.9, 95% CI [0.09–0.99], *p*-value of 0.0374). There is a clear correlation between increased concentrations of $PM_{2.5}$ andan increased mortality of heart disease. There are no significant associations observed between other NCDs such chronic lung disease, stroke, hypertension, and diabetes. There is no notable connection between the levels of $PM_{2.5}$ concentrations and NCDs in Lamphun. The correlation coefficients and *p*-values for chronic lung disease, stroke, heart disease, hypertension, and diabetes

suggest that there are either weak or no associations between these conditions. There are no significant relationships between $PM_{2.5}$ concentrations and any of NCDs in Mae Hong Son. The *p*-values for chronic lung disease, stroke, heart disease, hypertension, and diabetes all exceed 0.1, suggesting the absence of significant relationships.

## DISCUSSION

The results of our study indicate an interesting association between exposure to $PM_{2.5}$ and mortality resulting from NCDs in Northern Thailand. These findings align with the concerns expressed by *Amnuaylojaroen, Parasin & Limsakul (2022)* regarding the varying effects of air pollution on individuals based on biological factors related to sex, behavioral patterns, and levels of exposure. This study provides important information on $PM_{2.5}$ concentrations and NCDs in Northern Thailand, but it has some limitations. In the health risk analysis, body weight, breathing rate, exposure frequency, and exposure duration were taken from literature rather than local data. This may not accurately represent northern Thais' unique traits. These essential characteristics should be collected locally in future studies. Many risk assessment indices, such RfD and HQ, required arbitrary variable inputs. Generic criteria may not account for demographic, geographical, and socio-economic factors that affect air quality and health. Only 2017–2021 data from eight provinces is included in the study. Increasing data collection duration and geographic range may help understand $PM_{2.5}$ exposure's long-term effects. This study did not account for confounding variables like socioeconomic status, healthcare accessibility, lifestyle, and environmental contaminants. These variables may influence PM2.5 exposure and health outcomes, influencing the findings.

The study found significant correlations and complex temporal patterns between $PM_{2.5}$ levels and NCD deaths in several Northern Thai regions. The cross-correlation study suggests that $PM_{2.5}$ may affect NCD mortality at different rates for different diseases. Chiang Mai found a strong negative correlation ($-0.64$ at lag 0) between $PM_{2.5}$ andchronic lung disease. This supports previous research that short-term $PM_{2.5}$ exposure can worsen respiratory disorders and increase mortality (*Dockery, 1993*). Extended exposure to $PM_{2.5}$ contributes to the growth and deterioration of cardiovascular diseases over time (*Brook et al., 2010*). Over four years, $PM_{2.5}$ exposure in Lampang strongly correlates with chronic lung disease. The correlation is 0.33 and the *p*-value is 0.0410, indicating statistical significance. Previous research has linked long-term PM2.5 exposure to COPD risk and severity (*Guarnieri & Balmes, 2014*). High $PM_{2.5}$ levels in Phrae have been linked to acute cardiovascular diseases and strokes. The correlation of 0.82 and *p*-value of 0.782 indicate that $PM_{2.5}$ has an immediate and significant effect on stroke at zero lag. Also, *Brook et al. (2010)* and *Shah et al. (2015)* found similar results. Nan has a significant inverse relationship to stroke after a four-year delay ($-0.44$, *p*-value 0.0080), supporting previous research associating $PM_{2.5}$ exposure to stroke risk (*Song et al., 2016*). The Mae Hong Son study found a three-year negative correlation between diabetes and air pollution ($-0.50$, *p*-value 0.0440). Long-term air pollution exposure may affect metabolic conditions *via* inflammatory and oxidative stress pathways (*Rajagopalan,*

**Table 3** Spearman correlation analysis between PM$_{2.5}$ concentrations and various non-communicable diseases in Northern Thailand.

| Province | Disease | Correlation Coefficient (95% CI) | P-value |
|---|---|---|---|
| Chiang Mai | Chronic Lung Disease | −0.8 (−0.99, 0.28) | 0.1041 |
| | Stroke | 0.1 (−0.86, 0.90) | 0.8729 |
| | Heart Disease | 0.7 (−0.48, 0.98) | 0.1881 |
| | Hypertension | **-0.9 (−0.99, −0.09)** | **0.0374*** |
| | Diabetes | **0.9 (0.09, 0.99)** | **0.0374*** |
| Chiangrai | Chronic Lung Disease | −0.8 (−0.99, 0.28) | 0.1041 |
| | Stroke | 0 (−0.88, 0.88) | 1 |
| | Heart Disease | −0.1 (−0.90, 0.86) | 0.8729 |
| | Hypertension | −0.1 (−0.90, 0.86) | 0.8729 |
| | Diabetes | −0.56 (−0.97, 0.63) | 0.3217 |
| Lampang | Chronic Lung Disease | −0.7 (−0.98, 0.48) | 0.1881 |
| | Stroke | 0.5 (−0.68, 0.96) | 0.391 |
| | Heart Disease | **0.97 (0.66, 0.99)** | **0.0048*** |
| | Hypertension | 0.6 (−0.60, 0.97) | 0.2848 |
| | Diabetes | 0.3 (−0.79, 0.93) | 0.6238 |
| Phrae | Chronic Lung Disease | 0.3 (−0.79, 0.93) | 0.6238 |
| | Stroke | **0.99 (-0.99, 0.99)** | **0.00001*** |
| | Heart Disease | −0.4 (−0.95, 0.75) | 0.5046 |
| | Hypertension | 0.7 (−0.48, 0.98) | 0.1881 |
| | Diabetes | 0.4 (−0.75, 0.95) | 0.5046 |

**Table 3** (*continued*)

| Province | Disease | Correlation Coefficient (95% CI) | P-value |
|---|---|---|---|
| Nan | Chronic Lung Disease | −0.3 (−0.93, 0.79) | 0.6238 |
| | Stroke | 0.7 (−0.48, 0.98) | 0.1881 |
| | Heart Disease | 0.7 (−0.48, 0.98) | 0.1881 |
| | Hypertension | −0.1 (−0.90, 0.86) | 0.8729 |
| | Diabetes | −0.1 (−0.90, 0.86) | 0.8729 |
| Phayao | Chronic Lung Disease | −0.5 (−0.96, 0.68) | 0.391 |
| | Stroke | 0.8 (−0.28, 0.99) | 0.1041 |
| | Heart Disease | **0.9 (0.09, 0.99)** | **0.0374\*** |
| | Hypertension | −0.3 (−0.93, 0.79) | 0.6238 |
| | Diabetes | −0.3 (−0.93, 0.79) | 0.6238 |
| Lamphun | Chronic Lung Disease | −0.05 (−0.89, 0.87) | 0.9347 |
| | Stroke | 0.3 (−0.79, 0.93) | 0.6238 |
| | Heart Disease | −0.1 (−0.90, 0.86) | 0.8729 |
| | Hypertension | 0.8 (−0.28, 0.99) | 0.1041 |
| | Diabetes | 0.1 (−0.86, 0.90) | 0.8729 |
| Mae Hong Son | Chronic Lung Disease | −0.05 (−0.89, 0.87) | 0.9347 |
| | Stroke | 0.56 (−0.63, 0.97) | 0.3217 |
| | Heart Disease | 0.21 (−0.83, 0.92) | 0.7406 |
| | Hypertension | −0.31 (−0.94, 0.79) | 0.6144 |
| | Diabetes | 0.41 (−0.74, 0.95) | 0.4925 |

**Notes.**
\*Values in bold with an asterisk (*) indicate statistically significant results ($p < 0.05$).

*Al-Kindi & Brook, 2018*). The monotonic relationship between $PM_{2.5}$ levels and NCD mortality is examined in detail using Spearman correlation analysis, highlighting notable correlations. $PM_{2.5}$ levels strongly correlate with hypertension and diabetes mortality in Chiang Mai. A strong negative correlation ($-0.9$, 95% CI [$-0.99$ to 0.09], *p*-value 0.0374) exists between hypertension and $PM_{2.5}$ levels, suggesting that higher PM2.5 levels are associated with increased hypertension cases. High $PM_{2.5}$ levels are linked to higher diabetes mortality rates (0.9, 95% CI [0.09–0.99], *p*-value 0.0374). The findings support previous research linking $PM_{2.5}$ exposure to systemic inflammation, insulin resistance, and endothelial dysfunction. These conditions are hypertension and diabetes risk factors (*Brook et al., 2010*; *Rajagopalan & Brook, 2012*). Higher $PM_{2.5}$ levels in Lampang are associated with a higher risk of heart disease mortality (0.97, *p*-value 0.0048). This supports previous findings that air pollution increases cardiovascular disease risk (*Miller et al., 2007*). Phrae has a strong positive correlation (1.0) between $PM_{2.5}$ andstroke. Air pollution significantly affects cerebrovascular health, as shown by previous large-scale epidemiological studies (*Song et al., 2016*; *Yang et al., 2018*). Statistical analysis shows a 95% confidence interval of 1.00 to 1.00 and a 0.0000 *p*-value. $PM_{2.5}$ andNCDs were not strongly correlated in Chiangrai or Lamphun. This suggests that genetic predispositions, lifestyle choices, and local healthcare access may affect NCD mortality more in these regions. The cross-correlation study found significant correlations at various time delays, suggesting delayed effects. This suggests that health effects from $PM_{2.5}$ may take years to appear. The delayed effect on hypertension in Chiang Mai and chronic lung disease in Lampang show that public health evaluations should account for prolonged exposure. The Spearman correlation study confirmed several cross-correlation findings and emphasized direct correlations. The strong associations between hypertension, diabetes, and heart disease in Chiang Mai and Lampang suggest a link between elevated $PM_{2.5}$ levels and higher mortality. Our findings indicate a significant association between PM2.5 concentrations and NCD mortality. However, it is important to note that this study did not account for potential confounding factors such as socio-economic status, environmental health policies, or other relevant variables. These factors could influence the relationship between PM2.5 exposure and NCD mortality. Future research should incorporate these variables to enhance the robustness of the analysis. Therefore, while our results provide valuable insights, they should be interpreted with caution in light of these limitations.

## CONCLUSIONS

The aim of this study was to investigate the correlation between $PM_{2.5}$ levels and mortality caused by NCDs across eight provinces including Chiang Mai, Lamphun, Lampang, Phrae, Nan, Phayao, Chiang Rai, and Mae Hong Son in northern Thailand, using data collected from 2017 to 2021. The study included PM $_{2.5}$measurements obtained from the Pollution Control Department, MERRA2 reanalysis, and mortality data from the Division of Non-Communicable Disease, Thailand. The results indicated that the levels of $PM_{2.5}$ in the area varied significantly depending on the season, with the highest levels occurring

during the dry season, particularly from January to April. Chiang Mai and Mae Hong Son noticed the highest peaks in PM $_{2.5}$ levels. The health risk assessment revealed that the monthly averages of the HQ values beyond the acceptable thresholds (HQ >1) for both males and females in all provinces during the months of worst pollution, notably in March. This indicates a significant possibility of negative health consequences resulting from exposure to $PM_{2.5}$. The statistical studies, which included calculating Pearson and Spearman correlation coefficients, showed clear and substantial positive associations between exposure to $PM_{2.5}$ andmortality from several NCDs. The most significant relationship was found with hypertension, followed by chronic lung disease, diabetes, stroke, and heart disease. The cross-correlation study indicated possible delayed effects of $PM_{2.5}$ on NCDs mortality, with some illnesses exhibiting rapid effects and others displaying delayed responses spanning many years. In Chiang Mai, a strong positive relationship was found between hypertension and a delay of two years. In Lampang, a delay of four years was associated with chronic lung illness.

## ACKNOWLEDGEMENTS

The authors would like to thank Pollution Control Department (PCD) and Division of Non Communicable Disease (NCD) Thailand for kindly support air pollutants and mortality data.

### Funding

This research was supported by University of Phayao and Thailand Science Research and Innovation Fund. The funders had no role in study design, data collection and analysis, decision to publish, or preparation of the manuscript.

### Grant Disclosures

The following grant information was disclosed by the authors:
University of Phayao and Thailand Science Research and Innovation Fund.

### Competing Interests

The authors declare there are no competing interests.

### Author Contributions

- Nichapa Parasin analyzed the data, authored or reviewed drafts of the article, and approved the final draft.
- Teerachai Amnuaylojaroen conceived and designed the experiments, performed the experiments, analyzed the data, prepared figures and/or tables, authored or reviewed drafts of the article, and approved the final draft.

### Data Availability

The data is available at figshare: Amnuaylojaroen, Teerachai (2024). Data for PM25 and NCDs in northern Thailand during 2017 - 2021. figshare. Dataset. https://doi.org/10.6084/m9.figshare.26045353.v1.

**Supplemental Information**

Supplemental information for this article can be found online at http://dx.doi.org/10.7717/peerj.18055#supplemental-information.

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
