# Peer review of "Effect of PM2.5 on burden of mortality from non-communicable diseases in northern Thailand"

_PeerJ, doi:10.7717/peerj.18055_

## Round 0.1 · original submission · Major Revisions

Thank you for submitting your manuscript to PeerJ. We have obtained the reviews from four experts in their respective fields for your manuscript and they have left both some very detailed and helpful comments on different aspects of your work.

Based on the recommendations of the expert reviewers, we're recommending that major revisions are done to the manuscript. The reviewers have drawn our attention to the following aspects:

The statistical analysis
The reviewers have some concerns about your statistical methods, notably suggesting the use of a time-series analysis rather than focusing on correlations in the data, as they feel that the use of the Pearson correlation is not sufficient to support your conclusions.

Study design
One reviewer wishes to draw your attention to your study design being unsuitable to address potential confounding in your data.

Interpretations of results
The reviewers feel that some bias in the data, with how data from some specific stations were used, limits the robustness of the analysis and your ability to draw strong conclusions.

Clarifications of language used
There are some instances where the reviewers were unable to understand what you were refering to in the text, which has a knock-on effect of a lack of clarity in your methods. Please address these issues to ensure that your work is fully understandable and therefore replicable for a wider audience.

Citing the most relevant work in this domain for context
One reviewer has pointed out that your statement concerning a lack of research on the mortality from NCDs caused by PM2.5 may not, in fact, be accurate, and that several other studies (mentioned by the reviewer) are relevant to this work.

I invite you to carefully consider all of the comments from each of the reviewers and respond to all the points raised by sending a revised version of this manuscript to us.

Reviewer 1 ·

Basic reporting

no comment

Experimental design

1. The statistical analysis should be improved, e.g.. univariate models. Only using Pearson correlation is not sufficient to support the conclusions.
2. Please define dry season in the first time it is used. Could you please explain in more detail why Jun to December was not included in Table 3 analysis?

Validity of the findings

If my understanding is correct, mean observed PM2.5 concentrations in seven stations were used in the analysis. Could you please explain in more detail why the use of mean concentrations with different standard deviations is valid? For example, in Table 3, the standard deviation in March was 46.11 ug/m3.

Additional comments

1. Line 128, please remove “The” from “ET is exposure time (24 hour/day), and BW is body weight (kg), The these parameters were used”

·

Basic reporting

L.183 Figure 10 was mentioned but not attached. Most likely a repetition of L.180

Experimental design

No Comment

Validity of the findings

No Comment

Additional comments

L.124, Exposure Time(ET) seems not to be accounted for in the Average Daily Dose Formula stated

Interesting work done in finding the relationship between PM 2.5 exposure and NCDs in Northern Thailand. It will be great as a future work to look into the effects of other factors (such as age) on the
relationship between PM2.5 and death rates related to NCDs.

Reviewer 3 ·

Basic reporting

1. Figure 1 could be improved by providing additional information of each green polygon (what are these province?), and where major roadways are traveling from or major industry sites were located. Also some green polygons (that I assume is a unit of analysis) do not have a monitoring station. So how authors attribute exposures to these provinces?

2. Some languages were vague (e.g., describing QA/QC, Lines 102-110) and unnecessary (e.g., Lines 114 to 116).

Experimental design

1. In the introduction, authors claimed that “there has been little research on the mortality from NCDs caused by PM2.5,” which is not true as evidence is established – for instance, this review: Atkinson, R. W., Kang, S., Anderson, H. R., Mills, I. C., & Walton, H. A. (2014)). If talking specifically about Thailand, then Pothirat et al. (2018) and Mueller et al. (2021) have studied from both short-term and long-term exposure perspectives.
Ref: Pothirat, C., Chaiwong, W., Liwsrisakun, C., Bumroongkit, C., Deesomchok, A., Theerakittikul, T., ... & Phetsuk, N. (2021). The short-term associations of particular matters on non-accidental mortality and causes of death in Chiang Mai, Thailand: a time series analysis study between 2016-2018. International journal of environmental health research, 31(5), 538-547.);
Mueller, W., Vardoulakis, S., Steinle, S., Loh, M., Johnston, H. J., Precha, N., ... & Cherrie, J. W. (2021). A health impact assessment of long-term exposure to particulate air pollution in Thailand. Environmental Research Letters, 16(5))055018.

Authors need to provide a stronger rationale as to why their study contributes uniquely to the current literature.


2. Study design and statistical methods have major flaws and could not adequately answer the study question. With daily-level data, authors can opt for a more robust time-series analysis rather than studying correlations. The risk assessment indices (RfD, HQ, etc.) very much relied on arbitrary input of certain variables (such as variables in Table 2) and were not informative for a region with a diverse population (as authors described in lines 235-237). Even with the current choice of methods, authors should provide a measure of uncertainty (such as confidence intervals) for correlation coefficients. Also, considering how skewed the data might be, authors should pick a nonparametric measurement such as Spearman's correlation.

3. Lines 226 to 234 are the authors' biggest limitation, and it is not one that can be ignored. It greatly harmed the validity of current findings. The authors can totally change the study design to overcome this limitation, especially about confounding (things that change with PM2.5 and also correlate with outcomes), which greatly biased the coefficient measures. Authors should not use "effect" throughout. I wish the author could really consider redesigning the study.

Validity of the findings

My concerns lie in the exposure assessment, outcome attainment, and study design. Section 2 describes the major flaw in the study design (which is the biggest problem).

1. I have concerns about the exposure assessment part. First, excluding monitoring stations with more than 25% missing data will contribute to selection bias (for instance, stations in more polluted areas are likely to have broken/deteriorated instruments and, therefore, missing data). How many stations have this missing pattern? For less than 25% of missing, how did authors treat missing? Second, the authors provide no information on how exposure data were linked to mortality rates. Lastly, although QA/QC is mentioned, the actual procedures are not clearly described. Lines 102-110 do not provide details for readers to understand QA/QC.

3. How outcome (NCD mortalities) data was obtained is poorly described. Authors mentioned death rate, but provide no information on how this rate variable was calculated (what kind of rate – what is your denominator)? Authors also did not provide information on how each disease-related mortality was identified and what is the final unit of analysis.

6. Authors did not discuss their limitations in the discussion despite many, such as how exposure is representative of the area-level exposure, how confounding, and selection bias are not being adequately considered and controlled.

Additional comments

Summary of the study: In the present study, the authors investigated the association between daily average ambient PM2.5 exposure in 2020, measured by monitoring stations, and the prevalence of non-communicable diseases (NCDs) mortality rates in northern Thailand. Authors found significant positive correlations between PM2.5 and several NCD mortality rates.

Impression: Authors established the reason why this is an important topic to study, but failed to conduct a robust study with adequate design and methodologies. To me, the biggest problem is the study design itself that harmed the validity of results. Authors should work on selecting an adequate study design to answer their main research question and providing enough information in the methodology section.

Reviewer 4 ·

Basic reporting

The manuscript studies an interesting topic about the effect of PM2.5 on the burden of mortality from Non-Communicable Diseases (NCDs) in northern Thailand. This manuscript follows a conventional structure, which facilitates the comprehension of the study's aims, methods, results, and conclusions. The manuscript requires improvements in English language usage to enhance clarity and readability, as there are several grammatical errors and awkward phrasings. No raw data was provided. Additionally, the abstract does not contain a clear conclusion, which is necessary to summarize the key findings and implications of the study.

Below are some detailed comments:

Line 4: Please make sure you use the same font

Line 22: "increase the incidence and mortality of a Non-Communicable Diseases (NCDs)" should be "increase the incidence and mortality of Non-Communicable Diseases (NCDs)".

Line 22-23: You have included the acronym of NCDs twice.

Line 30: You should superscript the “3” in the unit ug/m3.

Line 205: Typo “hearth disease”; should be "heart disease". Also in Figure 5, Table 6, and Line 305

Reference: Please use a consistent reference format.

Figure 2:
The two lines have the same standard of 50 ug/m3, please correct this.
Figure 3:
Please provide labels on X and Y axis. Please include the unit on Y.

Table 4: Please spell out “Met. Factor”. You should add a column for “All” and see if there are correlations overall.

Table 6
Please provide p values related to Pearson’s Correlation.

Experimental design

The experimental design is generally well-structured. However, the use of Pearson correlation, the focus on short-term exposure effects, the single-year data, and the use of mortality as the health outcome metric raise several questions regarding the experimental design of this study. Here are some detailed comments:

The use of the Pearson correlation coefficient to analyze the relationship between PM2.5 and mortality from NCDs is a notable limitation. Pearson correlation only identifies linear relationships and cannot account for potential confounding variables such as age, socioeconomic status, smoking habits, and other environmental factors. This limitation may result in an oversimplified understanding of the complex interactions between PM2.5 exposure and health outcomes.

The focus on short-term exposure effects may overlook the cumulative and chronic impacts of long-term PM2.5 exposure. Long-term studies are essential to fully understand the chronic health risks and the development of diseases over time. Including a longitudinal component would provide a more comprehensive view of the health impacts. While short-term effects are important, they do not encompass the full spectrum of health risks associated with prolonged exposure. Long-term exposure to PM2.5 can lead to chronic health conditions, including hypertension, cardiovascular diseases, and respiratory diseases, which are not adequately addressed in a short-term study.

The study focuses on data from a single year (2020) and specific geographic locations in northern Thailand. This narrow scope may limit the generalizability of the findings. Expanding the temporal scope to include multiple years and considering data from a wider geographic area would provide a more robust and generalizable set of results.

The study primarily uses mortality as the health outcome metric. Including other health indicators, such as hospital admissions, incidence of new cases of NCDs, and quality of life measures, would offer a more comprehensive understanding of the health impacts of PM2.5 exposure.

Validity of the findings

The validity of findings is affected by the flaws in experimental design. See detailed comments in Section II Experimental design.

---

## Round 0.2 · Minor Revisions

Thank you for your resubmission. There are some very minor issues still outstanding. The reviewers have thanked you for including additional data to improve the robustness of your analysis.

Reviewer 1 ·

Basic reporting

N/A

Experimental design

N/A

Validity of the findings

N/A

Additional comments

1. lines 51-54: it is confusing to describe environmental factors and air pollution when the whole paragraph is particular about PM2.5.

2. please spell out the abbreviations the first time, e.g., COPD

3. line 305-312: this paragraph is more like a method section, not in the results section.

4. the study used NCD mortality instead of morbidity. Could the author explain why?

·

Basic reporting

no comment

Experimental design

no comment

Validity of the findings

no comment

Reviewer 4 ·

Basic reporting

No comment.

Experimental design

The authors have revised their manuscript to include data spanning from 2017 to 2022. This comprehensive update significantly enhances the robustness and depth of their final conclusions. By incorporating these additional years of data, the authors provide a more thorough analysis, offering greater insights to the validity of their findings.

The study used a variety of statistical analyses, including time series analysis, cross-correlation, and Spearman correlation, to investigate the associations between PM2.5 concentrations and mortality rates for various non-communicable diseases (NCDs). However, none of these analyses accounted for potential confounding factors that could influence these associations. Specifically, the study did not consider socio-economic factors, environmental health policies, or other relevant variables that might impact the relationship between PM2.5 exposure and NCD mortality.

Validity of the findings

No comment.

---

## Round 0.3 · accepted · Accept

Thank you, all reviewer comments have been dealt with